# Integrating (Nutri-)Metabolomics into the One Health Tendency—The Key for Personalized Medicine Advancement

**DOI:** 10.3390/metabo13070800

**Published:** 2023-06-27

**Authors:** Ionela Hotea, Catalin Sirbu, Ana-Maria Plotuna, Emil Tîrziu, Corina Badea, Adina Berbecea, Monica Dragomirescu, Isidora Radulov

**Affiliations:** 1Faculty of Veterinary Medicine, University of Life Sciences “King Mihai I” from Timisoara, Calea Aradului, No. 119, 300645 Timisoara, Romania; 2Faculty of Agriculture, University of Life Sciences “King Mihai I” from Timisoara, Calea Aradului, No. 119, 300645 Timisoara, Romania; 3Faculty of Bioengineering of Animal Resources, University of Life Sciences “King Mihai I” from Timisoara, Calea Aradului, No. 119, 300645 Timisoara, Romania

**Keywords:** metabolomics, nutrimetabolomics, One Health, public health, personalized medicine

## Abstract

Metabolomics is an advanced technology, still under development, with multiple research applications, especially in the field of health. Individual metabolic profiles, the functionality of the body, as well as its interaction with the environment, can be established using this technology. The body’s response to various external factors, including the food consumed and the nutrients it contains, has increased researchers’ interest in nutrimetabolomics. Establishing correlations between diet and the occurrence of various diseases, or even the development of personalized nutrition plans, could contribute to advances in precision medicine. The interdependence between humans, animals, and the environment is of particular importance today, with the dramatic emergence and spread of zoonotic diseases, food, water and soil contamination, and the degradation of resources and habitats. All these events have led to an increase in risk factors for functional diseases, burdening global health. Thus, this study aimed to highlight the importance of metabolomics, and, in particular, nutrimetabolomics, as a technical solution for a holistic, collaborative, and precise approach for the advancement of the One Health strategy.

## 1. Introduction

With recent world developments, the emergence of new diseases endangering global health creates particular challenges for the medical field. To meet these challenges, novel, highly precise technologies can be utilised to facilitate clinical investigations and identify therapeutic and preventive solutions.

In this paper, the term ‘(nutri-)metabolomics’ is used to refer to both metabolomics and nutrimetabolomics. Metabolomics, one of these novel technologies, is the analysis of the total profile of metabolites within a system (cell, tissue, or organism) in a certain time period and under certain conditions. It is an integral part of the ‘omics’ sciences and makes a direct link with the body’s phenotype, providing biochemical information in addition to genomic and proteomic data [1,2,3]. Metabolomic technology is constantly evolving and becoming widely used in an increasing number of fields, including agriculture, environmental chemistry, biotechnology, and, in particular, medical sciences for clinical diagnosis, toxicology, nutrition, drug progress, and health and disease management [3,4,5].

Nutrimetabolomics, or nutritional metabolomics, is an integral part of metabolomics with the goal of examining individual functional responses to different diets, analysing specific dietary biomarkers for targeted foods and diets, and investigating the interrelationship between risk factors for certain diseases and different diets both in the human and veterinary science fields [3,6,7]. Nutrimetabolomic technology aims to determine individual human and animal reactions to nutrition, as well as to identify and implement personalized nutritional plans for optimal health [3,8].

One Health considers the association of several decision-making authorities at the local, national, and global levels with the aim of establishing interdisciplinary collaborations to ensure optimal living conditions and health for people, animals, and the environment. The most recent diseases in animals and humans (SARS-CoV-2, Ebola, avian flu (H5N1), swine flu (H1N1), etc.) are examples of global issues and the increased vulnerability of humans, animals, and the environment to new disease outbreaks [9,10,11,12].

In this context, metabolomics can be used for the rapid elucidation of epidemiological conditions, disease pathology, therapeutic strategies, and prevention mechanisms for various diseases. Furthermore, the identification of specific individual biomarkers can lead to the development of personalized medicine (precision medicine), offering the possibility of controlling world health.

Thus, the purpose of this review was to summarize current knowledge from the period 2000 to 2020 on metabolomic and nutrimetabolomic research in the context of an interaction between human and veterinary medicine and to highlight the possible applications of these technologies in the One Health approach, to increase the scientific contribution to the advancement of a common public health strategy.

## 2. Metabolomics and Nutrimetabolomics

Given that nutrimetabolomics is a branch of metabolomics, in this study, one term or the other will be used, depending on the context, although the study refers to both technologies—metabolomics and nutrimetabolomics.

The term ‘metabolomics’ was introduced in the literature in 1998 and refers to the study of small molecules in a biological sample. In 2001, a group of biochemists founded the Society of Metabolomics [13], and the use of this technology in various fields of research has increased greatly since 2005 [14,15]. Among the ‘omics’ technologies (genomics, transcriptomics, and proteomics), metabolomics has become increasingly utilized, especially in the last decade. Metabolomics can be described as the overall analysis of small molecules in a biological fluid (blood, urine, culture broth, cell extract, etc.), produced or transformed in the body, as a result of the intervention of a stimulus (nutritional factors, stress, environment, drugs, etc.) [15]. All metabolites that make up the ‘metabolome’ represent the molecular fingerprint of an organism [16]. The profile of the metabolome (metabolic phenotype or ‘metabotype’) reflects the biological state of an organism (Figure 1) [16,17].

As can be observed from the numerous studies found on search platforms, metabolomics is a frequently applied technique. Interesting results have been obtained in the pharmaceutical (personal response to drug treatment, efficacy and/or toxicity of drugs), medical (biomarkers for disease diagnosis and prevention), and plant science fields (molecular structure, GMOs, etc.) [15,18].

Although metabolomics has recently been applied in nutrition, it is gaining increasing attention in this field [8]. Nutrition research has entered a new era, in which increased diet diversity and the relationships between nutrition and genotype, lifestyle, and diseases are investigated in great detail, with new methodological approaches, including ‘omics’ technologies to identify responses to various stimuli that have not been determined using traditional approaches [19,20]. Post-genomic technology also provides new methods and opportunities for research in the field of nutrition to explain individual differences in metabolism and assimilation of food and nutrients [15].

Nutritional fields combine traditional nutritional methods with genotyping and phenotyping, molecular epidemiology, and bioinformatics.

Nutrigenomics was the first nutrition domain to emerge investigating the effects of nutrients on genetic expression profiles and assessing how an individual’s genotype may influence nutrient uptake, excretion, or activity [15,21].

The next field that evolved was nutriproteomics, which analyses molecular and cellular changes in protein expression in response to nutrients and studies the interaction of proteins with nutrients in the food consumed [15,22].

Nutrimetabolomics is the dynamic and multivariate research of biological fluid or tissue responses to nutritional stimuli. This technology analyses the direct or indirect effect of diet on metabolism [15].

Nutrimicrobiomics refers to the genetic structure and functional capacity of microbial populations and the possibilities of the intestinal microbiome to process food and nutrients [15].

Food has long been known to influence the health and well-being of individuals, and nutrients have also been used as medicines to treat and prevent disease [16]. Research in this area has shown that diseases are often linked to poor or unbalanced nutrition [15,16,23,24]. At present, nutrition research has reached the level where metabolomics (nutrimetabolomics) is used to phenotype the nutritional status of individuals and facilitate the discovery of new biomarkers of specific nutrients or metabolic dysfunctions [15,25]. Moreover, metabolomic methods can be used to investigate the component metabolites of food in biological fluids and/or tissues, to study their bioavailability, the body’s response to a particular diet, and to analyse certain foods or nutraceuticals [16,26].

Food and nutrition science focus not only on the interrelationship between diseases and the consumption of macro- and micronutrients, but also study bioactive molecules that may be present in the diet in very small quantities [27]. These molecules can interact with various metabolic pathways and directly or indirectly modulate health, after being transformed by the intestinal microbiota [15]. Nutrimetabolomics offers the most accessible way to investigate the impact of nutritional interference on health, because the metabolic configurations of easily obtained biological fluids, such as blood, plasma, and urine, contain valuable information on both genetic heritage and environmental influences, including the contributions of dietary nutrients and their microbial transformations [15,28].

## 3. (Nutri-)Metabolomics in Human Medicine

Metabolomics developed rapidly in human medicine. It is primarily used to identify various biomarkers for the diagnosis, prevention, and treatment of various diseases, but also to investigate the role of the intestinal microbiota in the assimilation of dietary nutrients and to establish personalized nutrition plans or detailed food descriptions [15].

To identify the interconnections between lifestyle, diet, and health, it is necessary to establish specific biomarkers that provide information about nutritional profiles. In order to identify them, several studies have been launched to establish and develop nutritional markers [29,30] and to demonstrate their implications in the occurrence and development of metabolic diseases (obesity, diabetes), chronic diseases (HBP, CVD) or other diseases (cancer) [15,31,32,33,34,35,36,37,38].

In this regard, over time, the role of amino acids, fatty acids, or glucose metabolites in the onset and development of obesity or diabetes has been analysed [39,40,41,42]. It is known that the incidences of obesity/overweight and blood pressure (BP) risk values (prehypertensive/hypertensive) are globally widespread. A critical BP is a major risk factor for coronary heart disease and stroke, while obesity can, over time, lead to the onset of insulin resistance and type 2 diabetes or associated metabolic and cardiovascular diseases [15,43,44,45]. Various studies have been conducted on the influence of healthy diets on reducing the risk of developing cardiovascular diseases, type 2 diabetes, or various cancer types [15,46,47,48].

It is well known that nutrition is essential for maintaining life quality. Without nutrients, an organism cannot survive. Balanced nutrition can contribute to good health. Metabolomics is a key tool in modern nutritional research for analysing the number of calories, the ratio of food to fat, protein, and carbohydrate content, or the general intake of nutrients (Figure 2). Metabolomics is also used to analyse non-nutritive molecules that are apparently not vital to the body, but which, by their presence or absence, could affect the health and well-being of the body [16]. Thus, natural nutritive and non-nutritive molecules can be used as biomarkers for specific food consumption. These dietary or nutritional biomarkers can then be used to estimate the intake and quality of nutrients in the food consumed [15,24].

Biomarkers are usually specific to each nutrient, food compound, or complex food [15]. For example, recent studies have identified plasma trimethylamine oxide as a marker of fish consumption [49,50] or methylhistidines in urine as a marker of animal protein intake [51]. Another example would be the blood level of long-chain omega-3 fatty acids, identified as an indicator for seafood consumption. Alkylresorcinols can be identified as markers of whole grain consumption, whereas total plasma carotenoids can indicate fruit and vegetable intake [52,53,54,55]. Furthermore, proline betaine has been established as an indicator of citrus fruit consumption [56,57,58,59,60]. Such studies provide evidence that various compounds can be identified as food biomarkers, and their presence or concentration in biological fluids can be a tool to identify the diet consumed [15]. This type of analysis is new in that, by identifying nutrients (i.e., identifying foods consumed by establishing the metabolic profile) the opportunity is provided to evaluate different diets, to establish certain dietary patterns, and to objectively classify individuals, depending on the types of diets they consume [15,61].

Studies are underway regarding the role of intestinal microorganisms in nutritional status and human health. Metabolic changes between intestinal and host microbial populations, i.e., co-metabolism, can influence the metabolic profile of the host (metabotype), an aspect highlighted by studies in this field [62,63,64,65,66]. It has been shown that the intestinal microbiota can predispose or contribute to certain human diseases. Thus, establishing the quantitative profile of intestinal microorganisms may predict certain diseases by identifying metabolites or metabolic processes that will be affected at some point [67]. The identified metabolites may function as metabolomic biomarkers for assessing disease evolution and the body’s response to specific treatments. Considering the significant interindividual differences in the population of intestinal microorganisms and host-microbe interactions, individual variations in susceptibility to certain diseases can be identified, as well as the responses of each organism to different pharmaceutical or nutritional interventions. This underscores the need to develop nutritional modulation strategies through the creation and use of personalized diets, as well as individualized therapeutic regimens to increase healthcare efficiency [67].

The future goal of nutrition research is to assess the metabolic response of each individual to diet. Personalized nutrition is the process by which individuals will modify their lifestyle and diet according to the information they have about their current health or future disease predispositions [15,68]. The theory of personalized nutrition started from the need to apply specific nutritional recommendations for population groups with similar characteristics [15]. The division of subjects into groups first requires the determination of the metabotype of each individual and then, depending on their characteristic metabolic type and physiological state, a nutrition style that suits them is defined. Following this process, personalized nutritional plans can be designed to modify and control the diet for the benefit of health. Therefore, metabolomics is very important in establishing the nutritional phenotype of each individual and identifying specific biomarkers to characterize various nutrients (food metabolome) or different metabolic dysfunctions [15,69].

Recently, an increasing number of publications have emerged highlighting the connections between food science and other fields, such as agriculture, biology, medicine, genetics, or veterinary science [70,71,72,73,74]. Metabolomics is a technology that can facilitate interdisciplinary connections by associating health with food quality and nutritional value [15]. Foodomics is the determination of food components. Macro- (proteins, fats, or carbohydrates) and micronutrients (vitamins, minerals, or other molecular components) determine the nutritional value of edible compounds necessary to maintain human health [16,75,76,77]. Most of the food metabolome is composed of phytochemicals. The best-known phytochemicals, highlighted by metabolomics studies, are fruit polyphenols, tomato lycopene or soy isoflavones [16,70,75,78,79]. Metabolomics can also identify chemicals in the environment, such as insecticides, herbicides, fungicides, antimicrobials, toxins, and other contaminants, that may be present in food and that could be harmful to the health of individuals who consume it [16,80,81,82,83]. Metabolomic studies have also been performed to establish the effects of genetically modified plants on the quality and nutritional value of foods and their by-products [84,85]. The origins of foods can also be determined based on their metabolic profile [86,87]. Furthermore, metabolomic studies can be used to analyse the effects of production technologies applied to transform raw materials into finished food on their nutritional, biochemical, or sensory value [77,88,89]. In today’s society, these studies are becoming increasingly important, as consumers are increasingly aware and concerned about the quality and origin of the food they consume, as well as the effects of food processing on the nutritional value of food.

For both human and veterinary medicine, the use of metabolomic assessment to ensure food safety is of common interest. Moreover, studies have analysed and highlighted differences in metabolites between foods of animal origin contaminated with micro-organisms or parasites and uncontaminated foods [90,91,92]. Consumers are increasingly interested in organic food, mainly due to concerns about food quality and safety, as well as the publicity of the idea that organic products are healthier and safer than conventional foods. The relationship between different types of agriculture (organic vs. conventional) and the various metabolic components that have been identified through the use of diets of various origins in animal feed has attracted particular interest [15,93,94]. Thus, foodomics is important for ensuring public health on two levels: in the short term, it can be involved in clinical interventions to treat various metabolic dysfunctions, such as diabetes and obesity; and in the long run, it can be used in preventive public health strategies to prevent the occurrence of certain diseases [93].

## 4. (Nutri-)Metabolomics in Veterinary Medicine

The use of metabolomics in veterinary medicine is more limited compared with its use in the human medical field, in which the applications of this advanced technology have been intensively explored. Although the application of metabolomics in veterinary medicine is slightly behind that in human medicine, there has recently been a growing interest in this method, especially because it is non-invasive and only requires biological fluids, particularly for the investigation of animal health or disease (Figure 3) [95,96].

Human diseases are frequently studied by inducing the disease in laboratory animals, especially rodents [97,98]. It should be noted that the use of animals in the study of diseases is performed in compliance with the ethical principle of avoiding harm. In animal model studies, various pathological conditions are induced to understand the molecular phenomena associated with various diseases and their complications [98]. The veterinary field may offer the possibility to study some spontaneously occurring diseases in animals, even in parallel with the appearance of the disease in humans, both in terms of the disease phenotype and pathogenesis [97]. Metabolomics has been used to study a variety of common diseases, such as type 2 diabetes [99,100], different types of cancer [101,102], and congenital metabolic errors [97,103,104]. This technique has also been applied to the study of the effects of different medicinal regimens [105] or dietary interventions [6,106,107,108,109], toxins [110,111,112], and stress [98,101] on health.

Although the majority of metabolomic studies in veterinary medicine have focused on pets and primarily investigated the pathogenesis and diagnosis of various diseases, studies on farm animals have also been conducted [73]. This is due to the fact that pets and farm animals are increasingly being studied as a reference model for human diseases. In this context, increasing emphasis has been placed on precise comparative analyses of human and animal diseases to accurately characterize the disease pathobiology. The closest animal species to humans, from an anatomo-physiological and pathophysiological point of view, are dogs and pigs [113]. These species are frequently studied using metabolomic tools to obtain a more detailed perspective on the pathobiology of diseases in humans. These types of studies have proven to be very effective, due to the use of standardized environmental conditions in terms of animal feeding and housing [114,115,116].

However, some researchers argue that canine models would be most appropriate for metabolomic studies due to the similarities in physiology and pathophysiology between dogs and humans. Thus, canine models have high clinical significance for human medical research [117,118]. Canine subjects can be used in metabolomic research to identify specific biomarkers associated with many common diseases, including cancer, heart disease [119,120,121,122], liver disease [123], and parasitic or infectious diseases [124,125]. Furthermore, studies of common neuropsychiatric disorders have been performed using canine models [117,126]. Given the similarities in anxiety manifestation and symptoms in humans and dogs, canine subjects could be considered a suitable model for the study of human anxiety, thus contributing to the knowledge of disease mechanisms at the molecular level [117]. Dogs could also be helpful in the study of human ADHD, as it has been observed that dogs can spontaneously exhibit specific ADHD behaviours [126].

Studies of strictly veterinary medical interest using metabolomics have also been con-ducted—in particular, studies aiming to identify methods to increase the quality of life of animals and improve their health [95,96]. Furthermore, metabolomic studies in pets have evaluated differences in the metabolite profiles of dogs and cats [127] and identified specific metabolic fingerprints for various dog breeds [128,129,130] as well as specific toxicological biomarkers in dogs [131,132,133]. Moreover, the metabolomic profile of cerebrospinal fluid of healthy and epileptic dogs has been investigated [96,134,135].

Studies in nutrition physiology using metabolomics highlighted differences in the production of greenhouse gas precursors by establishing personalized diets for ruminants [136]. In the field of farm animal research, metabolomics can also be applied to investigate the effects of nutritional programming on genetics, i.e., monitoring the epigenetic effects of diet during early development [90,91]. For maximum efficiency of the genetic selection response related to the traits pursued, such as disease resistance, performance or product quality, the researchers identified precise biomarkers that can be used to predict the manifestation of phenotypic characteristics [137]. Thus, for metabolomic research in the field of genetics, using livestock species provides several benefits, compared with using human or laboratory animal populations, such as multi-generation genealogies, long-term selection lines with large phenotypic differences, routine population phenotyping, targeted mating, opportunities to standardize environmental conditions, and a genome organization very similar to that of humans—more similar than that of most laboratory animal species [73,116,138].

In livestock farming, the identification of the metabolomic profile can be a starting point for formulating future diets or specific treatments to increase animal production, animal welfare, and the quality of animal products for human consumption [139].

Nutrition is a complex process, conditioned by several factors. Nutritional metabolomics is a significant technology for studying the relationship between an organism and its diet, as well as the interactions between the organism and its genetics, lifestyle, or even gastrointestinal flora [8,127]. The development of pet nutrition science has led to an increase in the lifespan of dogs and cats and a significant improvement in their quality of life. Elucidating the interactions between the nutrient molecules consumed through diet and the pathophysiological mechanisms that can be encountered in the various systems of an organism provides the possibility to identify new intervention methods and clinical management of the patient [140]. In humans, the individual is considered the main element in the emergence of diversity in nutritional metabolomic research, due to interindividual differences in genetic background, age, sex, gastrointestinal flora, and lifestyle. In the case of pets, nutrition studies are less diverse, enabling the conduct of longer-term research in controlled environments with low variation and consistency in sample collection [127].

The body’s biofluids, such as blood, plasma, and urine, contain many metabolites in the form of small molecules, such as amino acids, lipid fractions, or sugars, some of which may function as specific biomarkers and a way to monitor pathological conditions or responses of the body to nutritional interventions [107,141]. Metabolomics can be a useful tool in understanding individual responses to different diets and developing personalized nutritional formulas to improve health [108,115,142,143]. In most studies, personalized nutrition refers to human medicine; however, in an increasing number of recent studies in the field of veterinary medicine, the theory of personalized medicine appears [95].

A future goal in the field of animal nutrition will be to develop metabolomic studies to understand the effects of applying personalized diets using systems biology approaches. This will rely on the interconnection of a large number of data resulting from biological processes involving nutrients and non-nutritive compounds present in various ingredients used in animal feed, to manage the production and/or health status of animals as needed [96].

## 5. (Nutri-)Metabolomics in the Context of One Medicine

The One Health concept is a global strategy that aims to expand interdisciplinary collaborations and communications in all aspects of health care for humans, animals and the environment [144].

The interconnection between human and veterinary medicine was outlined in the nineteenth century by the German pathologist, Rudolf Virchow (1821–1902). He said: ‘between animal and human medicine there are no dividing lines—nor should there be. The object is different but the experience obtained constitutes the basis of all medicine’ [145,146].

The One Health strategy was initiated in 2007 in collaboration with the American Medical Association and the American Veterinary Medical Association to defend, im-prove, and promote the health and well-being of all species by encouraging and supporting the teamwork of human physicians, veterinarians, and health researchers [147]. The concept of One Health is not new, but its importance has been increasingly recognized in recent years. A result of 21st century concerns, the One Health, One Medicine, One World theory is part of a broader set of research programs and policies, including biosecurity, food security, translational medicine, and global health, with the goal of removing interdisciplinary barriers. This is the scenario in which the future of the One Health initiative will be developed [148].

One Health aims to increase the quality of life of all species, human and animal, given that about two-thirds (60.3%) of emerging infectious diseases result from zoonoses, most of which originate in the wild (71.8%), according to the studies of Frank et al. in 2008 [149]. Karesh et al., in 2012, showed that over 60% of human infectious diseases are caused by pathogens common to animals [148,150]. In this regard, metabolomics can help elucidate specific metabolic pathways and identify in vivo methods of infection, as well as the mechanisms of action of pathogens, which could contribute to discovering novel preventive or therapeutic strategies [125].

Kafsack et al., in 2010, reported the main existing zoonoses, which included brucellosis, bovine tuberculosis, rabies, leptospirosis, human African trypanosomiasis, and malaria [124]. Other protozoan parasites that are the causative agents of serious infections in both animals and humans, including host-life-threatening diseases, are *Toxoplasma gondii* (toxoplasmosis), *Leishmania* spp. (leishmaniasis), *Cryptosporidium* spp. (cryptosporidiosis), and *Giardia* spp. (giardiasis). While these diseases are becoming less common in developed countries, they continue to endanger the health and lives of a significant number of the population in other parts of the world. Moreover, these pathogens infect other non-specific host species, such as insects, birds, or even mammals, to ensure their complete life cycle [124,151]. The metabolic adaptation of these pathogens can provide them increased resistance and allow them to exploit nutrient sources in variable niches. Detailed analysis of the metabolic profile of each developmental stage of infectious pathogens and the degree of adaptation and modulation of their metabolism to that of the host organism, as well as the host’s immune response to these infections, can ensure the development of new therapeutic and medicinal strategies for humans and animals [125].

From the perspective of One Health, One Medicine, the study of zoonotic diseases is particularly important, given that it involves the inevitable and complex interaction of linked biological systems: humans, animals, and the environment [124]. The biological system comprises all living organisms and their relations with the environment. Thus, metabolomics research cannot be viewed singularly; all the functional systems of an organism are correlated with each other and with the environment. Thus, a complex analysis of metabolites and a multidisciplinary interpretation are needed to understand the overall functioning of organisms. For example, the analysis of plant metabolomics cannot be separated from that of mammals, because plant matter is the main source of food and nutrients for animals. Studies cannot only focus on understanding how the human body works, because, through the nutrients we consume from food of animal and vegetable origin, our biology is closely related to that of plants and animals. Moreover, studies cannot focus in particular on multicellular organisms when they are in direct coexistence with unicellular ones (e.g., gut microbiota).

The One Health approach must also include non-communicable diseases, which, according to the World Health Organization, affect the lives of over 36 million people each year [152]. Among the non-communicable diseases common to humans and animals, investigated by numerous studies, we can mention cardiovascular diseases, coronary heart disease, hypertension, obesity, diabetes, various types of cancer, epilepsy, etc. [124,153,154,155]. Because both humans and pets (dogs and cats) can be affected by the same diseases, research on diseases in one species may be useful in the early intervention for other affected species [124]. Determining the metabolic profile (metabolomics/metabonomics), compared to other ‘omics’ profiles, has the advantage of mirroring the entire ecosystem of the body, which provides an overview of the functionality of the whole organism. These data are still difficult to analyse, especially at the level of an individual, due to the diversity of consumed food and thus the wide range of nutrients that enter the body. Due to this inevitable complexity, metabolomics research has sought to focus on the use of animal models whose genetic profile and environmental conditions can be more easily monitored compared with those of the human population [15].

Since the 20th century, when canaries were first used as carbon monoxide detectors in coal mines to prevent poisoning in miners, animals have been commonly used as sentinels for various pathologies, to maintain public health [156]. The use and monitoring of sentinel animals and the collection of all data on the occurrence of diseases in animal populations help to identify sources of disease, analyse the effectiveness of drugs or preventive intervention schemes, highlight the epidemiology of pathogens, or design an early intervention plan, thereby ensuring public health [157,158].

Epidemiological research in the veterinary field has several advantages over epidemiological research conducted in humans, such as a shorter period of disease development, an easier process to obtain necropsy and histopathological data, and lower costs. Unlike other types of laboratory studies (e.g., cell cultures), sentinel animal studies allow a more accurate study of conditions manifested in humans [159,160].

The One Health trend aims to support and encourage comparative studies on diseases that can be found in both humans and animals, the most common being obesity, diabetes, autoimmune disorders, and cancer [161,162]. Studies that use metabolomics to study these diseases aim to understand disease pathophysiology and identify biomarkers or therapies to improve public health.

Much of the research that has used sentinel animals to elucidate common pathologies has focused on the study of different types of cancer in pets, especially dogs, which live very close to humans, intimately sharing the same environment and consuming approximately the same categories of food. This is because most cancers found in dogs have been shown to have about the same pathological characteristics, biological basis, clinical manifestations, proportional morbidity, and identified risk factors (including eating habits), as those found in humans [158]. Thus, metabolomic or nutrimetabolomic studies involving detailed analyses of cancer in dogs complement current knowledge in human medicine on the efficacy of immunotherapy, prolonged release of drugs, gene therapy, and the clinical-pathological picture of this ubiquitous disease [163,164]. In the future, these types of ‘omics’ studies will allow the identification of personalized treatments and contribute to the development of personalized medicine in both humans and animals.

Recently, there has been a continuous increase in the average body weight among the human population, resulting in an increase of obesity. As the link between the occurrence of obesity in humans and pets is proving to be increasingly close and complex, much more than what was known in the past, obesity should be seen as a unique health issue and addressed in the context of One Health [154,165]. Through a psychological analysis, obesity in both dogs and their owners could be seen as an involuntary transfer of attitude towards the feeding process or eating habits of owners to their dogs and as an orientation towards the humanization of pets [165,166]. Nutrimetabolomic studies in obese animals can explain the conditions of obesity establishment and its pathophysiology, so as to prevent associated diseases, such as heart disease or diabetes.

As humans and pets share the same environment, the same ecosystem, studies in which animals can be used as sentinels for humans are very appropriate; however, there are instances when humans can also serve as sentinels in some animal health circumstances [167]. Given that metabolomic studies are more advanced in human medicine, the latest medical discoveries can also be applied in veterinary medicine, for the benefit of animal health, helping to identify gaps in the control of animal diseases.

‘Omics’ technologies are fundamental in advancing One Health knowledge. The application of these advanced technologies, together with a multidisciplinary collaboration, are the key tools in improving not only human and animal health, but also food safety and security and, implicitly, ecosystem health [168].

Food safety refers to the preparation, handling and storage of food using methods that ensure the prevention of foodborne illness [169]. Foodborne illnesses are serious problems that can threaten public health. The number of cases of foodborne diseases, including those caused by waterborne pathogens, is increasing [146]. Infectious diseases and food safety are interrelated. Animal disease influences animal production and, thus, the availability of food of animal origin for human consumption. In addition, food insecurity and malnutrition are aggravating factors for opportunistic infections [148]. Thus, plant and animal food safety is an area in which metabolomics can be successfully used, for identifying specific biomarkers of food contamination [170].

The World Food Summit described food security as ‘when all people, at all times, have physical, social, and economic access to sufficient, safe and nutritious food’ [148,171]. In addition to quantity, food quality is also an extremely important aspect of food security. In this context, dogs and cats can serve as sentinels to identify possible food contamination, including chemical contamination, or to determine the effects of consuming genetically modified plants and their derived food and feed products [148,172]. Therefore, in the context of One Health, ‘plant health’ must also be seen as an integrated aspect, especially since plant metabolomics is a frequently studied field [173,174].

Whereas in less developed countries there are people who suffer from hunger due to lack of food, in developed countries there are widespread micronutrient deficiencies, also known as hidden hunger [175]. Recent research in the field of nutrition and the implementation of its results in public health has been shown to have an important effect on reducing the incidence of diseases caused by deficiencies of essential nutrients. However, metabolic disorders are not only caused by deficiencies of essential nutrients but also by micronutrient or non-essential nutrient deficiencies. The increasing frequency of metabolic changes in energy regulation, manifested by diabetes, obesity, or atherosclerosis, has affected a large part of the world’s population, including citizens of the world’s most developed countries [115,176,177]. Nutrimetabolomics can be used for the early identification of the effects of nutrient deficiencies on the human body. These studies can be performed on animals first, as they are cheaper and less time consuming [178].

The usual methods of nutritional analysis that focus on assessing the link between diet and health, in general, do not cover the detailed understanding of the interdependence between a single nutrient and disease occurrence. Using such a perspective, and specific analyses, unique disease biomarkers can be identified and used for disease prevention. To determine the metabolic health of individuals, a more precise approach to nutritional assessment is needed, and this is possible using nutrimetabolomics [115].

Nutrient deficiencies in humans are closely linked to the availability of micronutrients in food (animal or plant origin), and this is dependent on the availability of micronutrients in animal feed and soil (for plants). However, the presence of micronutrients in food sources is not enough; the body’s ability to metabolize some nutrients is also essential because metabolism is closely correlated with health. These interdependencies are part of the concerns of the One Medicine, One Health concept (Figure 4).

The holistic approach of the One Health concept remains one of the main ways to study the direct and underlying causes of food insecurity, malnutrition, and poor health and maximize human, animal, and environmental well-being. The added value of the One Health approach is that the benefits to human health are achieved through simultaneous investigations into human and animal health.

The future of mankind depends on a symbiotic relationship between humans, animals, and the environment. The rapid evolution of the planet towards industrialization, highly advanced technologies, and massive urbanization with a tendency towards human domination has led to an ecosystem imbalance and an increased vulnerability of human health. These trajectories have put humanity in unprecedented situations, with a rapid multiplication of emerging and re-emerging infectious diseases, as shown by the 2019 SARS-CoV-2 pandemic, amplified by the establishment of antimicrobial resistance and associated with an exponential increase in noncommunicable diseases.

In these situations, precise and rapid tools are needed to intervene in the face of threatening challenges to public health. In this context, the integration of metabolomics in most research areas will contribute to identifying customized solutions that will help restore the symbiosis between the human, animal, and plant populations and the environment, under the comprehensive “umbrella’’ of the One Health strategy.

## 6. Conclusions

This review provides an overview of the applicability of metabolomics and nu-trimetabolomics to the study of various health-threatening situations and in the development of innovative solutions to prevent global health threats. In human medicine, metabolomic research can contribute to the identification of specific biomarkers with life-saving importance for various critical diseases, leading to great advances in personalized medicine. In the veterinary field, these advanced technologies can help improve health, welfare, animal husbandry, and food security. The integration of metabolomics and nutrimetabolomics in human and veterinary medical research will help eliminate the barriers that still exist between animal health and public health control measures and support the progress of the One Health strategy.

## 7. Future Directions

Nutrimetabolomics, within the context of One Health, is a crucial field that explores the impact of nutrition on the interconnected health of humans, animals, and the environment. This emerging discipline has the potential to revolutionize personalized nutrition and healthcare. Looking ahead, several trends will shape the development and application of nutrimetabolomics.

Advancements in metabolomic technologies will enable more comprehensive and accurate profiling of small molecules and metabolites in the body. This will lead to the identification of new nutrient-metabolite biomarkers for early detection, monitoring, and assessment of nutritional status and disease risk. The intricate interactions between the gut microbiome and dietary components will be explored, allowing for personalized recommendations based on an individual’s unique microbial composition. Personalized dietary recommendations will be generated by analyzing an individual’s metabolomic data, including genetic variations and lifestyle factors, to optimize health and prevent chronic diseases. Integration of nutrimetabolomics with nutrigenomics will provide a comprehensive understanding of the interplay between genetics, metabolism, and nutrition, leading to personalized dietary interventions based on an individual’s genetic makeup. Comparative studies across different species will uncover commonalities and differences in nutrient metabolism, contributing to the development of nutritional interventions that benefit multiple species and promote overall health within One Health ecosystems. Nutrimetabolomics will play a role in the prevention and control of zoonotic diseases by understanding their transmission between animals and humans and guiding the development of nutritional interventions. Predictive models incorporating nutrimetabolomic data, along with other clinical and lifestyle factors, will assess an individual’s disease risk, allowing for targeted preventive strategies. Nutrimetabolomics will guide the development of nutraceuticals and functional foods tailored to an individual’s metabolic profile, optimizing therapeutic outcomes. Natural alternatives to antibiotics in animal and human health will be discovered through nutrimetabolomics research. Nutrimetabolomics will assess the impact of environmental pollutants on metabolic health in humans and animals. To fully realize the potential of nutrimetabolomics in personalized medicine within the One Health framework, integration into routine clinical practice is crucial. Collaboration and data sharing among researchers, healthcare professionals, veterinarians, and environmental scientists will enhance knowledge exchange and enable evidence-based interventions. Comprehensive databases of nutrimetabolomic data from different species and ecosystems will facilitate research capabilities, while artificial intelligence and machine learning algorithms will help analyze vast datasets, extract meaningful patterns, and generate personalized recommendations.

## Figures and Tables

**Figure 1 metabolites-13-00800-f001:**
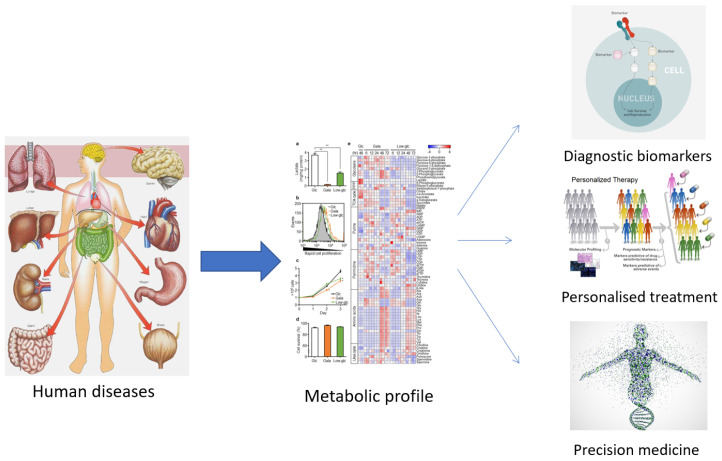
The usefulness of establishing the metabolic profile for medicine.

**Figure 2 metabolites-13-00800-f002:**
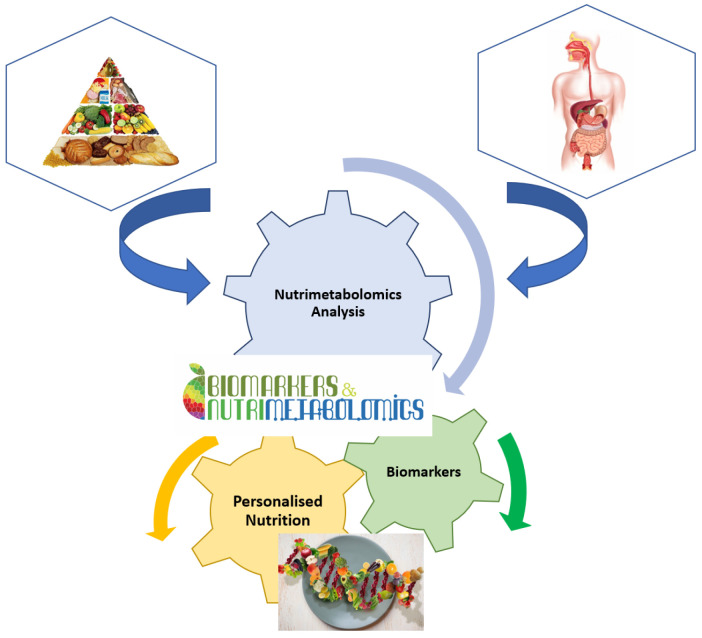
Implication of nutrimetabolomics in nutrition scrience.

**Figure 3 metabolites-13-00800-f003:**
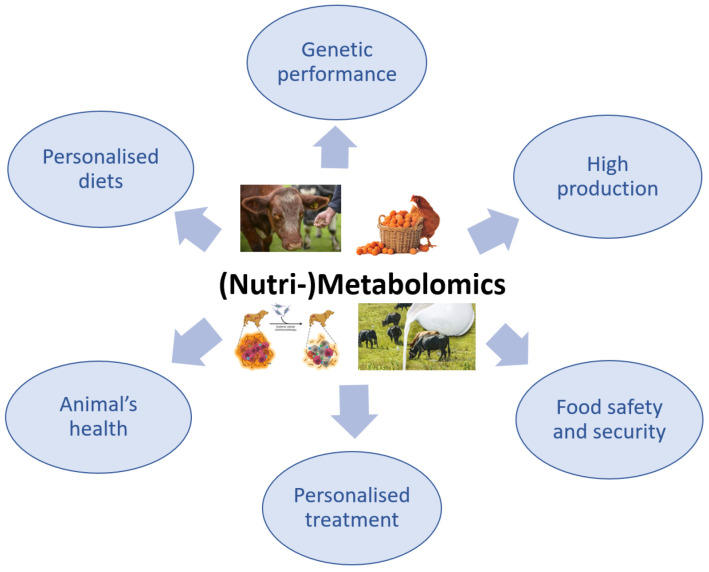
The importance of (Nutri-)Metabolomics in veterinary science.

**Figure 4 metabolites-13-00800-f004:**
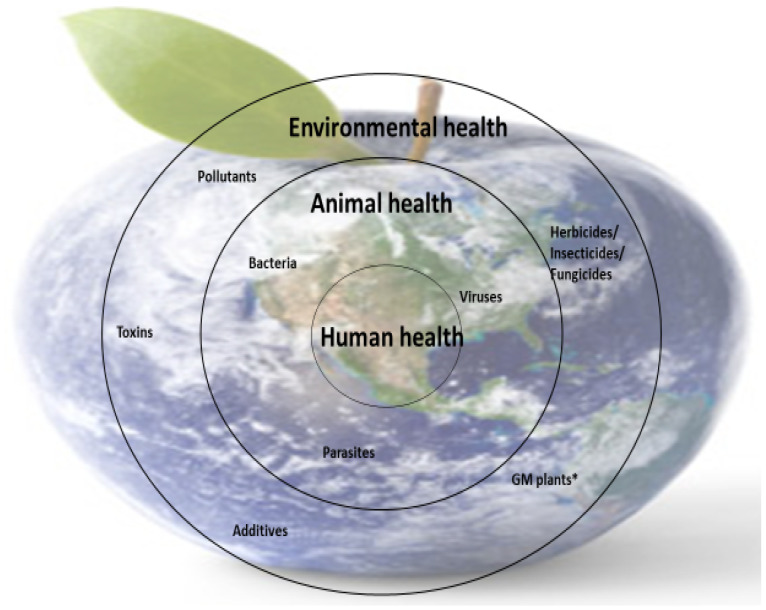
Schematic representation of the importance of One Medicine. * GM plants—genetically modified plants.

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
