# Peer review of "Integrating (Nutri-)Metabolomics into the One Health Tendency—The Key for Personalized Medicine Advancement"

_metabolites, 2023, doi:10.3390/metabo13070800_

Round 1
Reviewer 1 Report
Reviewed is the manuscript “Integrating (Nutri-)Metabolomics into the One Health tendency – the key for personalized medicine advancement” submitted by Ionela Hotea, et. al.
Summary of the Manuscript
The manuscript offers a thorough analysis of the use of metabolomics and nutrimetabolomics in veterinary and human medicine. The authors talk about how these cutting-edge technologies could be used to find precise biomarkers for various diseases, improve animal health, and raise animals' quality of life. The concept of One Health, a global strategy to increase interdisciplinary cooperation and communications in all facets of healthcare for people, animals, and the environment, is also emphasized in the study.
Strengths
· The manuscript is well-structured and provides a detailed overview of the subject matter.
· The authors have done a commendable job in highlighting the importance of metabolomics in both human and veterinary medicine.
· The paper effectively emphasizes the concept of One Health and its relevance in the context of metabolomics.
· The authors have provided a comprehensive list of references, indicating a thorough literature review.
Weaknesses
· The manuscript could benefit from a more detailed discussion on the statistical methods used in metabolomics studies.
· The paper could provide more specific examples of studies where metabolomics has been successfully applied.
· The authors could have discussed more about the limitations and challenges in the application of metabolomics and nutrimetabolomics.
Major Recommendations
· The authors should consider providing a more detailed discussion on the statistical methods used in metabolomics studies. This would be particularly beneficial for readers who are not familiar with these methods.
· The paper could benefit from more specific examples of studies where metabolomics has been successfully applied. This would help to illustrate the practical applications of this technology.
Minor Recommendations
· The authors should consider discussing more about the limitations and challenges in the application of metabolomics and nutrimetabolomics. This would provide a more balanced view of the subject matter.
· The authors could consider including a section on future directions in metabolomics research. This would provide readers with an idea of the potential developments in this field.
· The manuscript could benefit from a more detailed discussion on the statistical methods used in metabolomics studies. The authors should consider discussing the types of statistical analyses that are commonly used in these studies, such as multivariate analysis, cluster analysis, and principal component analysis. Additionally, the authors could discuss the challenges in analyzing metabolomics data, such as dealing with high-dimensional data and issues related to multiple testing.
Overall, the manuscript provides a comprehensive and insightful review of the application of metabolomics and nutrimetabolomics in human and veterinary medicine. The authors have effectively highlighted the importance of these technologies in identifying specific biomarkers for various diseases and improving animal health. The emphasis on the concept of One Health is particularly commendable, as it underscores the interconnectedness of human, animal, and environmental health. However, the manuscript could benefit from a more detailed discussion on the statistical methods used in metabolomics studies, as well as more specific examples of the practical applications of these technologies. Despite these minor shortcomings, the manuscript is a valuable contribution to the literature on metabolomics and nutrimetabolomics.
Author Response
Thank you for your appreciation.
- Statistical analysis of metabolomics data is not a subject of this review. This topic is already covered in other review articles where the concept of metabolomics is discussed in general.
- The examples of metabolomics studies are endless. In this review, we have chosen to present the most common applications of metabolomics and nutrimetabolomics, with relevance to the One Health context.
A future review article summarizing most of the possible applications in metabolomics may be an excellent idea.
- A section on future directions in (nutri-)metabolomics research in the One Health approach have been added.
Reviewer 2 Report
The article discussed the importance of metabolomics, particularly nutrimetabolomics, in advancing personalized medicine and the One Health approach. The study emphasized the potential applications of metabolomics in nutrition research, such as identifying biomarkers of specific nutrients, evaluating diets, and establishing personalized nutrition plans. The authors also highlighted the importance of utilizing advanced technologies to facilitate clinical investigations and identify solutions to global health challenges. Integrating metabolomics and the One Health approach can lead to advances in precision medicine, contributing to optimal living conditions and health for people, animals, and the environment. Overall, the paper provided valuable insights into the applications of metabolomics in health research, emphasizing the importance of interdisciplinary collaborations, precision medicine, and advanced technologies in addressing global health challenges.
However, there are only two references from 2021 onwards. It would be advisable for the authors to incorporate any relevant advancements from the past two years.
Author Response
Thank you for your appreciation.
The study period was 2000 to 2020, so the review cites articles from that period. We have added this observation in the text at the purpose of the review.
Reviewer 3 Report
Review work: Integrating (Nutri-)Metabolomics1 into the One Health ten- 2 dency – the key for personalized medicine advancement
A very interesting review. I have no comments
Author Response
Thank you for your appreciation.